# Accelerating ultrafast magnetization reversal by non-local spin transfer

Quentin Remy [1], Julius Hohlfeld[1], Maxime Vergès[1], Yann Le Guen[1], Jon Gorchon [1], Grégory Malinowski[1], Stéphane Mangin [1] & Michel Hehn [1] ✉

When exciting a magnetic material with a femtosecond laser pulse, the amplitude of magnetization is no longer constant and can decrease within a time scale comparable to the duration of the optical excitation. This ultrafast demagnetization can even trigger an ultrafast, out of equilibrium, phase transition to a paramagnetic state. The reciprocal effect, namely an ultrafast remagnetization from the zero magnetization state, is a necessary ingredient to achieve a complete ultrafast reversal. However, the speed of remagnetization is limited by the universal critical slowing down which appears close to a phase transition. Here we demonstrate that magnetization can be reversed in a few hundreds of femtoseconds by overcoming the critical slowing down thanks to ultrafast spin cooling and spin heating mechanisms. We foresee that these results outline the potential of ultrafast spintronics for future ultrafast and energy efficient magnetic memory and storage devices. Furthermore, this should motivate further theoretical works in the field of femtosecond magnetization reversal.

Ultrafast demagnetization[1], the subpicosecond quenching of magnetic order resulting from ultrafast heating of electrons via various external stimuli[2–5], is seen as an opportunity to push spintronics and magnetic data storage devices toward the subpicosecond regime[6]. Because the external ultrashort stimuli bring the system strongly out of equilibrium[7], the resulting magnetization dynamics is purely longitudinal[8,9] therefore circumventing limitations due to precessional dynamics[10]. A fundamental limitation for magnetization switching, however, is the critical slowing down (CSD) of magnetization dynamics: when the amplitude of the magnetization reaches zero, the rate of change of magnetization tends toward zero[11–15]. This CSD is a universal feature that exists close to phase transitions[16], a ferromagnetic/paramagnetic transition in our case.

To explain CSD and how it can be avoided, we present a simple mean-field two-level model of magnetism based on the widely accepted picture, in the context of ultrafast magnetism, that the spin angular momentum contributing to magnetization is mostly localized on atomic sites[5,9,11–15,17–20]. In this model, diagrammatically represented in Fig. 1, each atomic spin in the system has a spin quantum number of 1/2, and both levels are separated by an energy $\Delta E = 2mk_B T_C$, due to the

mean field created by the other spins, where $m = n_\downarrow - n_\uparrow$ is the normalized magnetization (with an opposite sign compared to angular momentum), $n_{\uparrow(\downarrow)}$ is the density of spins on the spin-up (down) level and $T_C$ is the Curie temperature. Independently of any external environment, we define a spin temperature $T_S$ in a microcanonical fashion, valid at all times and even when the spin system is out of equilibrium (see Supplementary Notes). It depends on the normalized, out-of-equilibrium, magnetization via:

$$T_S = \frac{mT_C}{\tanh^{-1}(m)}, \tag{1}$$

consistent with the standard equilibrium result. Decreasing the magnetization implies populating the level of higher energy which increases disorder and hence the spin temperature. Because of the conservation of angular momentum (or energy), magnetization can only change if the spin system is coupled to at least one external system with temperature $T$. The spin system is out of equilibrium if $T_S \neq T$. This leads to the general temperature model description of ultrafast demagnetization[1] except that here the spin system is not

[1]Université de Lorraine, Institut Jean Lamour, UMR, 7198 CNRS Nancy, France. ✉e-mail: michel.hehn@univ-lorraine.fr

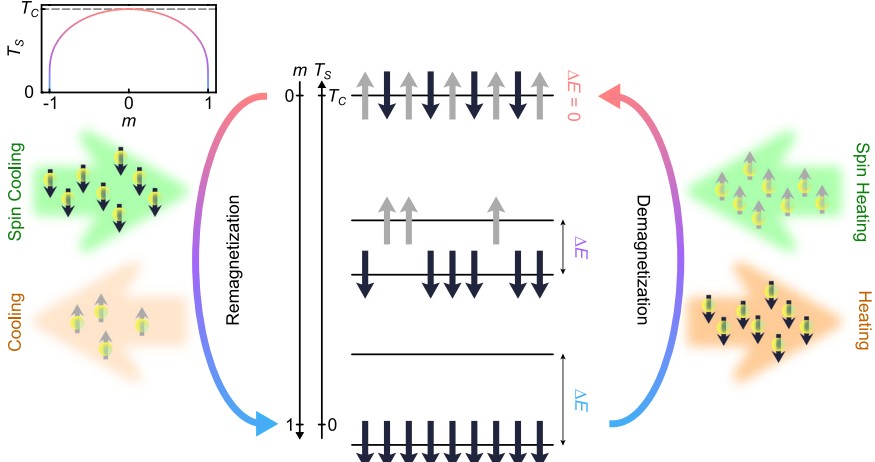

**Fig. 1 | Two-level model of ultrafast magnetization dynamics.** A system with magnetization $m$ formed by localized spins can be represented by the population of its different energy levels. Here the two levels can be populated by up (gray arrows) and down (black arrows) spins (center of the figure). To different populations correspond different energy splittings $\Delta E = 2mk_B T_C$ in the mean-field approach considered. For each magnetization and its corresponding possible configurations, one can associate a spin temperature $T_S$ (see inset). Due to the coupling of the (two-level) spin system with its external environment, there can be an exchange of angular momentum which will either cool down the spin system (remagnetization) or heat it up (demagnetization) depending on the polarization of the transferred angular momentum. If the coupling with the external environment is strong enough and if the latter undergoes ultrafast changes, the resulting magnetization dynamics may be ultrafast as well, i.e., in the absence of CSD. The standard way to obtain ultrafast magnetization dynamics is to heat up the system (energy transfer), leading to an ultrafast dissipation of angular momentum. It is not possible to achieve a similarly fast remagnetization in general because cooling of the system relies on heat dissipation via the sample substrate on a much longer timescale, and CSD may happen around $m = 0$. Spin cooling and spin heating work analogously but via angular momentum transfer. However, contrary to standard cooling and heating, spin cooling and heating can be achieved interchangeably by changing the sign of the external angular momentum. Because an external source of angular momentum (with constant sign) can lead to magnetization reversal (and not spin dissipation), spin heating becomes spin cooling when $m$ crosses 0 (see inset) and vice versa.

necessarily in equilibrium[12,20–22]. Magnetization dynamics[20,21] $dm/dt = 2(W_{\uparrow\to\downarrow} n_\uparrow - W_{\downarrow\to\uparrow} n_\downarrow)$ is characterized by an asymmetric change in the transition rates $W_{\downarrow\to\uparrow}$, from the spin-down level to the spin-up level, and $W_{\uparrow\to\downarrow}$, from the spin-up level to the spin-down level[21,22]:

$$\frac{W_{\downarrow\to\uparrow}}{W_{\uparrow\to\downarrow}} = e^{-\frac{\Delta E}{k_B T}}. \tag{2}$$

Magnetization dynamics can then only be launched by changing $T$ and the spin system reaches equilibrium when the ratio of populations in both levels is equal to the ratio of transition rates $n_\uparrow/n_\downarrow = W_{\uparrow\to\downarrow}/W_{\downarrow\to\uparrow}$. According to Eq. (2), the equilibrium spin system is described by a Boltzmann distribution at temperature $T_S = T$ for $T \le T_C$. When $T_S$ approaches the Curie temperature, the populations of each level become identical. Thus, magnetization can only change if both transition rates are different. However, one sees that in this case $\Delta E = 0$ and it follows that $W_{\downarrow\to\uparrow} = W_{\uparrow\to\downarrow}$ independently of the value of $T$, consistently with the fact that at zero magnetization, both spin levels are degenerate. This implies that the magnetization will remain at zero even though the equilibrium value of the normalized magnetization $m_{eq}$ corresponding to the lowest free energy value, and given as a solution of $m_{eq} = \tanh(m_{eq} T_C/T)$, is not zero. Cooling of the spin system via transfer (actually dissipation) of angular momentum in the external bath is then prohibited even though the completely quenched state is unstable[16]. An external source of angular momentum is required to break the rotation symmetry of the paramagnetic state and lift the degeneracy of both spin levels. In more realistic situations, magnetic fluctuations, which are neglected in the mean-field approach, will allow the system to retrieve some finite magnetization, but this process is much slower than ultrafast demagnetization[11]. If there is now a non-zero supply of angular momentum from the external bath, for instance, due to spin-polarized electrons[20,21], the principle of detailed balance (Eq. (2)) becomes:

$$\frac{W_{\downarrow\to\uparrow}}{W_{\uparrow\to\downarrow}} = e^{-\frac{\Delta E + \Delta E_{ext}}{k_B T}}, \tag{3}$$

where $\Delta E_{ext}$ is the energy splitting induced by the external source of angular momentum (later simply referred to as external source of angular momentum). As long as $\Delta E_{ext} \ne 0$, the magnetization dynamics is no longer frozen (actually even when $\Delta E + \Delta E_{ext} = 0$, see Supplementary Notes) and CSD can be overcome. In this case, magnetization dynamics and spin temperature, are not only affected by $T$, but by $\Delta E_{ext}$ as well.

Hence, we define the notion of spin cooling and spin heating of the magnetization as a decrease and respective increase in the spin temperature $T_S$ due to an external source of angular momentum $\Delta E_{ext}$. This is to be compared with (normal) cooling and heating of magnetization, i.e., the decrease and respective increase in the spin temperature $T_S$ due to energy transfer via changes in the temperature $T$ of an external bath. Close to $m = 0$, when $\Delta E$ is much smaller than $\Delta E_{ext}$:

$$\frac{W_{\downarrow\to\uparrow}}{W_{\uparrow\to\downarrow}} \approx e^{-\frac{\Delta E_{ext}}{k_B T}}, \tag{4}$$

the sign of $\Delta E_{ext}$ dictates which transition rate dominates, and spin cooling and spin heating can be used to drive magnetization in either direction no matter what the external temperature is. The external source of angular momentum therefore permits an analogous of population inversion in the standard two-level model while heating can at best make both populations equal, at the Curie temperature. This is the reason why the speed of magnetization dynamics in response to conventional heating or cooling is always limited by CSD, whereas ultrafast spin cooling and spin heating allow complete and ultrafast control of magnetization dynamics around $m = 0$.

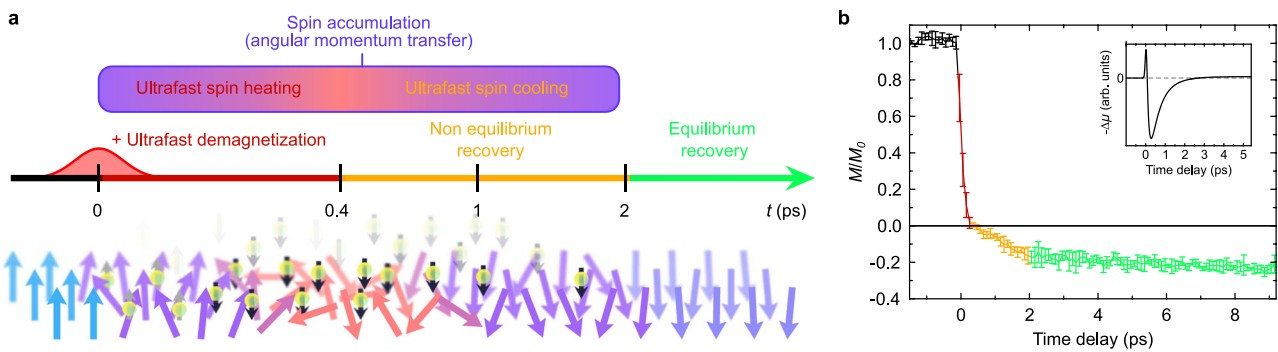

**Fig. 2 | Mechanism and temporal evolution of magnetization reversal of the free [Co/Pt] layer in our collinear. a** Schematic of the magnetization reversal process. Solid arrows represent the ferromagnet (atomic) spins, while bullets with arrows represent the injected spin accumulation. The red-shaded Gaussian curve represents the 100 fs (800 nm) pump laser pulse. **b** Experimental results of the ultrafast magnetization reversal of the [Co/Pt] ferromagnetic multilayer (P configuration) as obtained from our TR-MOKE microscopy measurements and analysis. The magnetization dynamics is seen to happen in three steps: ultrafast demagnetization with spin heating, nonequilibrium recovery of magnetization with ultrafast spin cooling, and equilibrium recovery of magnetization. Error bars are defined in the Supplementary Information. The inset shows the predicted spin accumulation generated in the free [Co/Pt] multilayer by the GdFeCo alloy.

Such spin cooling and spin heating of magnetization can be observed when the external source of angular momentum is a magnetic field[23], spin-polarized hot electrons[24–26], the spin angular momentum coming from one sublattice in ferrimagnets[27–29] or a Ruderman–Kittel–Kasuya–Yosida (RKKY) exchange coupling[30,31]. In these cases, however, magnetization was either far from quenched or CSD was still present (see the supporting information in ref. [28] for a manifestation of CSD in ferrimagnets).

In this work, we directly investigate how magnetization reversal can be accelerated, and, therefore, how CSD can be avoided, using spin-polarized hot electrons as the external source of angular momentum. We observe the magnetization dynamics of a ferromagnetic [Co/Pt] multilayer subjected to a spin current, the latter being generated by the ultrafast demagnetization of a GdFeCo alloy[32]. It has been shown that this permits magnetization reversal of the ferromagnetic multilayer[33–35] and is, therefore, the perfect playground to study magnetization dynamics around $m = 0$. We first show that magnetization reversal happens in around 400 fs, being the fastest magnetization reversal for a ferromagnet ever observed. Magnetization then reaches its equilibrium value in only 2 ps, corresponding to a full reversal considering the sample temperature at this instant. We explain these accelerated dynamics by a combination of ultrafast spin heating before reversal, allowing demagnetization with less heating of the sample itself, and ultrafast spin cooling after reversal, preventing CSD. Using the bipolarity of the external spin current[32], we show that magnetization can be reversed back only 650 fs after it was originally reversed. Finally, we directly observe large spin cooling and spin heating, changing the magnetization by up to 30% of its room temperature value in the first few picoseconds of the dynamics, and show how it prevents CSD. We substantiate these claims using a model that explicitly computes the transition rates in Eq. (3) when $\Delta E_{\text{ext}}$ is (minus) the spin accumulation $\Delta \mu$ of conduction electrons[20], i.e., the difference between the chemical potential for spin-up and spin-down electrons. We show that we can qualitatively reproduce our ultrafast reversal of magnetization as well as spin heating and spin cooling. We conclude from our mean-field two-level model and simulations that the speed of the dynamics is only restricted by the amount of angular momentum supplied by the external bath.

## Results

### Subpicosecond magnetization reversal
To study the ultrafast magnetization dynamics of [Co/Pt] when it is subjected to a spin current, we used a spin-valve structure based on previously studied spin-valve model systems[33–35] namely: Sapphire (Substrate) / Ta(5) / Pt(4) / [Co(1)/Pt(1)]$_2$ / Co(0.6) / Gd$_{33}$Fe$_{60,3}$Co$_{6,7}$(5) / Cu(10) / [Co(0.6)/Pt(1)]$_2$ / Ta(5) where the numbers between parentheses represent layer thicknesses in nanometers. A schematic overview of the sample is displayed in Supplementary Fig. 2, together with its measured magnetic properties. The multilayer of interest, whose magnetization is monitored, is the top (free) [Co(0.6)/Pt(1)]$_2$. We measured its magnetization dynamics using time-resolved complex magneto-optical Kerr effect microscopy as detailed in "Methods" and Supplementary notes. We use the other (pinned) magnetic [Co(1)/Pt(1)]$_2$ / Co(0.6) / Gd$_{33}$Fe$_{60,3}$Co$_{6,7}$(5) multilayer as the spin current source and its engineered properties are discussed in "Methods" and Supplementary notes. As previously shown in ref. [32], this spin current, and the spin accumulation it generates[36], has a bipolar structure, as shown in the inset of Fig. 2b and Supplementary Fig. 11a. A first peak corresponds to a spin polarization dominated by angular momentum coming from the FeCo magnetic sublattice of GdFeCo, while the more predominant second peak, of opposite sign, corresponds to a spin polarization dominated by angular momentum coming from the Gd sublattice. We call P a configuration where the magnetization of the free [Co/Pt] multilayer is initially parallel to the magnetization of the FeCo sublattice. The other configuration is called AP. We now only focus on the top [Co(0.6)/Pt(1)]$_2$ multilayer subjected to this external source of angular momentum.

We first focus on the P configuration, which was previously shown to lead to magnetization reversal of [Co/Pt] due to the second peak of the spin current[33–35]. The experimental results, shown in Fig. 2b, exhibit three steps: (1) a complete ultrafast demagnetization from 0 to 400 fs; (2) a fast recovery in the opposite direction to around 25% of the initial amplitude from 400 fs to 2 ps; (3) a slow recovery after 2 ps (see Supplementary Fig. 10d for longer time delays) as the sample cools down. We explain this result via the mechanism sketched in Fig. 2a. During step (1), magnetization is rapidly quenched due to ultrafast heating. At first, the ultrafast demagnetization is slowed down for a short period of time due to ultrafast spin cooling of the first peak of the spin current (gray arrows in Fig. 2a) but is then accelerated due to the ultrafast spin heating of the more predominant second peak of the spin current (black arrows in Fig. 2a). Because of this external source of angular momentum, no CSD of magnetization dynamics happens and the magnetization reverses at 400 fs, following the direction set by the second peak of the spin current. Because the spin current is expected to last for at least 1 ps, ultrafast spin cooling is expected to happen during this second step of the dynamics. This explains the absence of a plateau where magnetization stays at zero for around 2 ps in the case of reversal due to RKKY coupling[31]. However, similarly to the case of a reversal with RKKY coupling, magnetization then saturates at around 25% of its room temperature value. This is because the sample is still

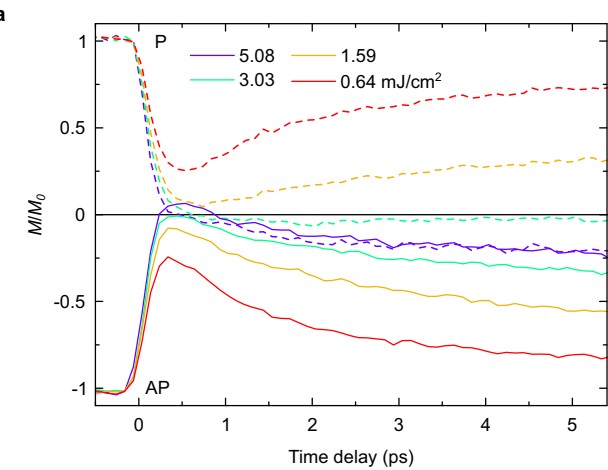

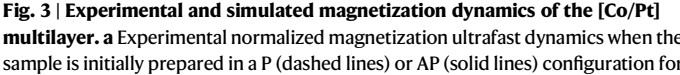

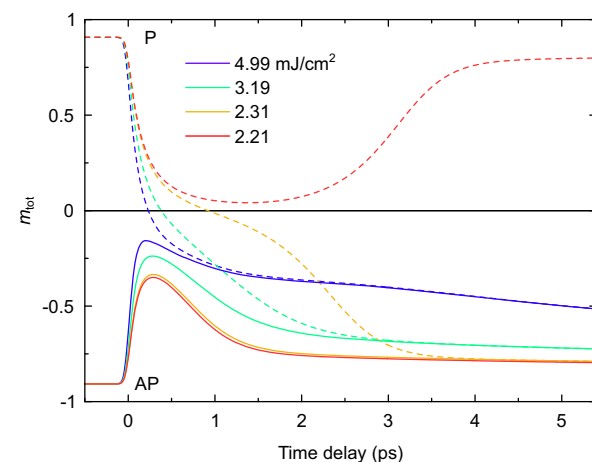

**Fig. 3 | Experimental and simulated magnetization dynamics of the [Co/Pt] multilayer. a** Experimental normalized magnetization ultrafast dynamics when the sample is initially prepared in a P (dashed lines) or AP (solid lines) configuration for four different fluences as indicated. **b** Results obtained from the simulations where $m_{\text{tot}}$ includes the normalized magnetization coming from both localized and itinerant electrons in the s-d model.

hot a few picoseconds after laser irradiation. In step (3), the magnetization barely changes, consistently with the fact that samples typically need more than a nanosecond to go back to room temperature. Also, note the typically relatively low Curie temperature of [Co/Pt] multilayers[37]. We also observe a small influence of the external magnetic field used to reset magnetization on a longer timescale (see Supplementary Fig. 10d), as already studied in detail previously[23]. Because of the change of trend of the magnetization dynamics at 2 ps, we hypothesize that the magnetization is at equilibrium (with the electrons) starting at this time delay.

The out-of-equilibrium dynamics of the mean-field two-level model introduced previously when it is exposed to spin-polarized conduction electrons has been derived in ref. [20]. We combine this model with a two-temperature model and a spin accumulation generated by the ultrafast demagnetization of GdFeCo to simulate our results, as detailed in "Methods" and Supplementary notes. Note that here the spin accumulation generated by the ferromagnetic multilayer itself is neglected. It is interesting to see what this model predicts when magnetization approaches zero. In the presence of a spin accumulation, the dynamics is governed by the following equation:

$$\frac{\mathrm{d}m}{\mathrm{d}t}\Big|_{m=0} = \frac{-\Delta\mu}{2\tau k_B T_C} \propto -\Delta\mu \qquad (5)$$

Where $\tau \propto 1/(k_B T_C)$ is the characteristic time for the transfer of spin angular momentum between itinerant and localized electrons. This shows that magnetization regrows in a given direction depending on the sign of the spin accumulation. The proportionality constant essentially depends on how efficiently the angular momentum is transferred from the external bath to the spin system[20,38]. In the absence of a spin accumulation, one finds:

$$\frac{\mathrm{d}m}{\mathrm{d}t}\Big|_{m=0} = \frac{m^2 T_C}{3\tau T_e} \propto \frac{m^2 T_C^2}{T_e} \qquad (6)$$

Where $T_e$ is the temperature of the conduction electron bath. In this case, one retrieves CSD, independently of the electronic temperature, as previously explained, and magnetization always tends toward zero.

The results of our simulations are shown in Fig. 3b (see also Supplementary Fig. 11). They show that our model can qualitatively reproduce the ultrafast magnetization reversal behavior as explained in "Methods" and Supplementary Information. The two-step recovery

of magnetization observed in Fig. 2b is clearly reproduced (see also Supplementary Fig. 11e), consistently with our claim that magnetization is at equilibrium after 2 ps (see Supplementary Note). Noticeably, our model overestimates the amplitude of the reversed magnetization at a short timescale. This can be corrected by slightly changing the Curie temperature in our model but will not change qualitatively the observed behavior governed by Eqs. (5) and (12) (see "Methods").

Following ref. [31], we define the reversal time of magnetization as the duration it takes for the amplitude of magnetization to cross zero. Magnetization, in our case, then reverses in around 400 fs. Another definition is the duration required for magnetization to recover its initial room temperature amplitude. Although the latter definition seems more natural, it strongly depends on how the total system cools down to room temperature and is mostly a heat dissipation optimization problem. A better indication of the speed of ultrafast magnetization reversal due to an ultrashort stimulus, also more relevant for practical applications, is the time delay between two such stimuli to reverse magnetization twice consecutively. For GdFeCo, this delay has been measured to be 300 ps[39], 7 ps for GdCo[40], and 10 ps in MnRuGa[41]. We consider this magnetization reswitching or rewriting, caused at high temperatures by a fast sign change of the spin current that originates from heating by a single laser pulse, in the next section.

## Ultrafast magnetization rewriting

Figure 3a shows the experimental magnetization dynamics of the [Co/Pt] ferromagnetic multilayer when it is initially prepared in one of the two configurations (P and AP). At low fluence, the magnetization dynamics in both configurations are similar and show the usual ultrafast demagnetization behavior[42]. It is worth noting that at a fluence of 3.03 mJ/cm² in the P configuration, close to the estimated static threshold fluence for single pulse reversal, the magnetization barely changes anymore once it reaches zero. This evidences an additional cause of CSD not taken into account by our model, which we believe is due to complex magnetic textures[11]. In the AP configuration, we see that at high fluence, the magnetization still seems to have the standard ultrafast demagnetization dynamics behavior, except that it transiently crosses 0 between 200 and 850 fs (see Supplementary Fig. 10b, c). Note that as shown in Supplementary Fig. 10e and 10f, the pinned layer (GdFeCo) is only demagnetizing and is barely influenced by the spin current originating from the ultrafast demagnetization of the free layer (Co/Pt), consistently with static measurement of threshold fluences[35] (see "Methods").

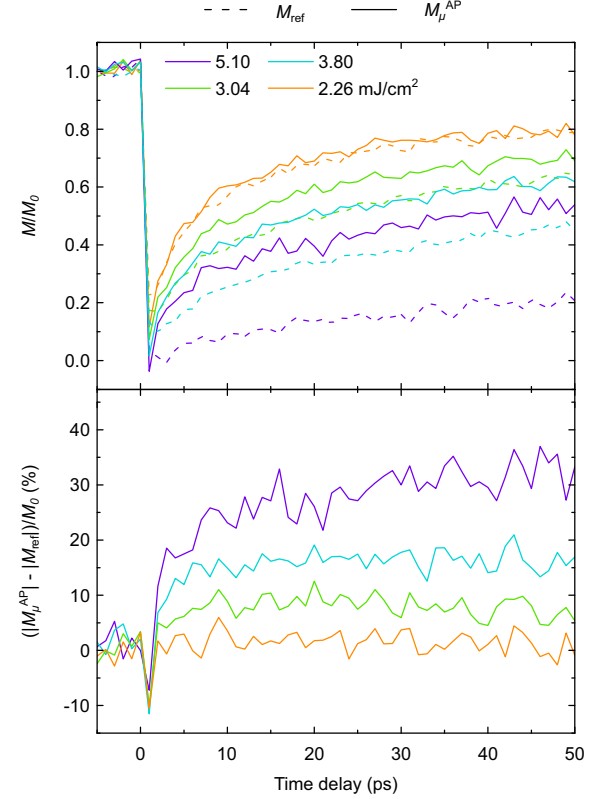

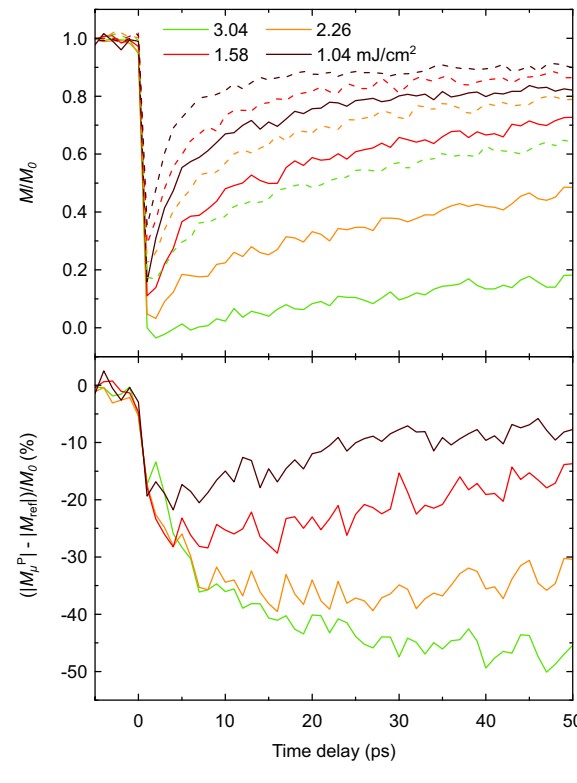

**Fig. 4 | Experimental observation of ultrafast spin cooling and heating in the [Co/Pt] ferromagnetic multilayer. a** The top panel compares the magnetization dynamics in the reference sample $M_{ref}$ (dashed line) with the magnetization dynamics of the free [Co/Pt] ferromagnetic multilayer in the spin-valve sample in the AP configuration $M_\mu^{AP}$ (solid line). The bottom panel shows the effect of the spin current generated by the GdFeCo layer on the magnetization dynamics of the free [Co/Pt], i.e., $(|M_\mu^{AP}| - |M_{ref}|)/M_0$. **b** The top panel compares the magnetization dynamics in the reference sample $M_{ref}$ (dashed line) with the magnetization dynamics of the free [Co/Pt] ferromagnetic multilayer in the spin-valve sample in the P configuration $M_\mu^P$ (solid line). The bottom panel shows the effect of the spin current generated by the GdFeCo layer on the magnetization dynamics of the free [Co/Pt], i.e., $(|M_\mu^P| - |M_{ref}|)/M_0$. The color code for the considered fluences is shown in the top panel in each case. The time delay step has been chosen to be larger and of 1 ps. Zero time delay is here chosen to be the longest delay where there is no observable signal.

In order to understand the magnetization dynamics in the AP configuration, the bipolarity of the external spin current is needed. If the observed transient reversal was solely due to error bars (see Supplementary Fig. 10c) and the magnetization was actually completely quenched between 200 fs and 850 fs, we should observe CSD, which is not the case. Rather, the magnetization recovers in the original direction and follows exactly the same magnetization dynamics as in the P configuration (highest fluence in Fig. 3a) for time delays greater than 1 ps, where it is mostly driven by the system temperature. This suggests that the first peak of the spin current causes the transient reversal while the much stronger (delayed) second peak forces the magnetization to recover. Magnetization dynamics around $m = 0$ (when $\Delta E \ll \Delta E_{ext}$) can then be completely manipulated by an external source of angular momentum. The transient reversal in the AP configuration is not reproduced by our simulations for our choice of parameters but an impact of the bipolarity of the spin current can still be observed, as is evidenced by the behavior of the magnetization dynamics at high fluence shown in Supplementary Fig. 11d.

**Ultrafast spin cooling and heating**

To further study the role of the spin current, a Ta(5) / Pt(4) / Cu(10) / [Co(0.6)/Pt(1)]$_2$ / Ta(5) structure, referred to as the reference sample, was studied where no external angular momentum is provided to the ferromagnetic [Co/Pt] multilayer. We thus compare the magnetization dynamics of the reference sample (where there is no spin injection), with those of the ferromagnetic layer within the spin valve, for both AP and P configurations, and therefore two opposite spin population

injections. For a low laser fluence of 2.26 mJ/cm², we retrieve almost the same magnetization dynamics in the reference sample and in the spin valve in an AP configuration, as shown in Fig. 4a. Only a slight difference in the dynamics at very short delays is observed between the samples, which we attribute to spin currents, similar to observations in ref. [24]. At higher fluence, however, the magnetization amplitude in both samples start to deviate significantly at long delays. The bottom panel of Fig. 4a shows the evolution of the normalized difference between the magnetization dynamics of both samples (which we attribute to the presence or absence of spin current) as a function of pump-probe delay. At the highest fluence, a first negative peak to around −10% is observed, followed by a jump of around 30% in 2 ps (to reach 20%). The difference saturates after 10 ps and can reach around 30%. This can be understood by the fact that CSD of magnetization is avoided due to ultrafast spin heating (first jump) and ultrafast spin cooling (second jump) ($\Delta E_{ext} \neq 0$). For longer time delays ($\Delta E_{ext} = 0$), magnetization recovers much slower in the reference sample because of CSD, while in the spin valve, ultrafast spin cooling avoids the CSD so that the magnetization recovery can follow due to standard cooling. The external magnetic field plays only a role on a longer timescale, as seen in Supplementary Fig. 10d. Similarly, in Fig. 4b, the magnetization dynamics of the reference sample is compared to the magnetization dynamics of the spin valve starting from a P configuration. In this case, we study lower fluences which do not allow the free layer to reverse its magnetization. In this way, the second peak of the spin current is always expected to cause spin heating of the magnetization. Indeed, as shown in the top left corner of Fig. 1, spin heating becomes spin

cooling when magnetization crosses zero. The laser power is the same as in Fig. 4a but even at the smallest fluence of 1.04 mJ/cm$^2$, we can see a clear difference between the magnetization dynamics with and without the presence of the spin current. The normalized difference between both kinds of magnetization dynamics is also plotted in Fig. 4b and reveals the spin heating effect of the spin current on magnetization at early times. The difference between magnetizations at long delays are this time significantly higher, reaching −50% at the maximum fluence, not causing magnetization reversal of the free layer. This large difference compared to Fig. 4a is reproduced by our model (see Supplementary Fig. 11f).

Even if our model is able to qualitatively reproduce the measured ultrafast reversal of the [Co/Pt] ferromagnetic layer, including the two steps recovery of magnetization above the reversal threshold fluence, a more quantitative description would require considering the spin accumulation generated by [Co/Pt], the lifting of the mean-field approximation[11,17], going beyond the s-d model[43], and a more accurate description of the energy and angular moment transport in heterostructures in general[20,36].

In conclusion, we experimentally observed that magnetization reversal of a ferromagnetic layer is achievable in 400 fs using one single femtosecond laser pulse. We demonstrate that this fast reversal is happening thanks to an external source of spin angular momentum, which allows to overcome the CSD observed near phase transitions. This spin current has a bipolar profile evidenced by the observation of magnetization reswitching in 650 fs. This accelerated magnetization dynamics is explained in terms of ultrafast spin cooling and ultrafast spin heating of magnetization, which can be directly observed by comparison with magnetization not subjected to this spin current. We believe that these results will encourage further work to enable more practical energy-efficient ultrafast spintronic devices to be imagined, such as the possibility of reversing the magnetization of a ferromagnet due to the ultrashort spin current generated by another ferromagnet.

## Methods
### Sample growth
The samples with structures Sapphire / Ta(5) / Pt(4) / [Co(1)/Pt(1)]$_2$ / Co(0,6) / Gd$_{33}$Fe$_{60,3}$Co$_{6,7}$(5) / Cu(10) / [Co(0,6)/Pt(1)]$_2$ / Ta(5) and Glass / Cu(10) / [Co(0,6)/Pt(1)]$_2$ / Ta(5) were grown using DC/RF magnetron sputtering with a base pressure of around $10^{-7}$ Torr.

### Static measurements
The first characterization of the samples is performed with static measurements. Hysteresis loops have been measured with the TR-MOKE microscopy setup, as shown in Supplementary Fig. 5, with the pump blocked. In order to separate the behavior of both magnetic layers in the spin-valve sample, we performed hysteresis loop measurements of both the spin valve and the reference sample using standard continuous wave MOKE as shown in Supplementary Fig. 1. From these two measurements, we can deduce that the [Co/Pt] multilayer has the smallest coercive field (around 5.7 mT) while the ferrimagnetic layer (coupled to another [Co/Pt] multilayer) has a coercive field of around 35 mT.

The threshold fluences required to reverse a given magnetic domain were obtained by fitting the magnetic domain size vs pulse energy graph (Supplementary Fig. 4), assuming that the laser beam has a Gaussian profile. More details about this standard procedure can be found, for instance, in ref. [35]. In these measurements, a 35 fs linearly polarized laser pulse (800 nm) is irradiated on the top [Co(0.6)/Pt(1)]$_2$ side of the sample while static (not time-resolved) MOKE microscopy is performed on the other side of the sample. We found a threshold fluence of $2.19 \pm 0.05$ mJ/cm$^2$ for the magnetization reversal of the single [Co/Pt] multilayer and a threshold fluence of $5.39 \pm 0.11$ mJ/cm$^2$ for the magnetization reversal of the GdFeCo/[Co/Pt] multilayer.

The threshold fluences obtained from time-resolved measurements are susceptible to deviate from these values as multiple pulses are required for stromboscopique experiments. A typical static measurement where the laser pulse fluence is higher than both threshold fluences is shown in Supplementary Fig. 3. The fit required to obtain threshold fluences is shown in Supplementary Fig 4. The RGB color associated to a pixel with a given gray value is given by the lookup table "phase" of the open-source software ImageJ.

### TR-MOKE microscopy
In contrast to previously investigated spin valves[33–35], a (thicker) ferromagnetic [Co/Pt] multilayer is added at the bottom and exchange coupled to the GdFeCo layer to pin it in a given direction. This is verified by the typical major and minor hysteresis loops shown in Supplementary Fig. 2b, revealing two independently switching layers where the film with the small (large) coercivity is the free (pinned) layer (see Supplementary Fig. 1). The four distinct remanent states (named P(AP) when the transition metal magnetizations of the free and pinned layers are parallel (anti-parallel) with a superscript + (-) when the free layer points up (down)) can be observed at zero field. Depth-sensitive time-resolved magneto-optical Kerr effect microscopy is used to measure the magnetization dynamics in both P and AP configurations as a function of the pump fluence. Measurements are performed under a constant magnetic field H+ or H− to reinitialize the magnetization of the top [Co(0.6)/Pt(1)]$_2$ multilayer between each pulse of the pump-probe measurement. The only purpose of the additional [Co(1)/Pt(1)]$_2$ / Co(0.6) multilayer (in brown in Supplementary Fig. 2a) is to increase the coercive field of the GdFeCo layer and therefore simplify the reinitialization of the magnetization during our dynamics measurements.

The TR-MOKE microscope that we used is summarized in Supplementary Fig. 5 and follows the method of ref. [44]. A chopper with diffusing "Magic™ tape" (Scotch®) was used to destroy the coherence of the probe laser beam and thus eliminate laser speckle and interference for the imaging. The laser pulses with an original full width at half maximum (FWHM) duration of 100 fs and wavelength of 800 nm were generated by a Yb femtosecond fiber laser with regenerative amplifier with a repetition rate set to 50 kHz. An optical parametric amplifier was used to change the wavelength of the probe beam. The pump and probe wavelengths were 800 nm and 850 nm, respectively. Both the pump and the probe pulses irradiate the top [Co(0.6)/Pt(1)]$_2$ side of the sample, as shown in Supplementary Fig. 2. A combination of half-wave plate and polarizers was used to change the laser powers. The probe laser power was set to ~40 mW and the pump laser power was varied depending on the experiment (60 mW and 120 mW for the reference and spin-valve samples, respectively). At the sample position, the polarization was p (s), the measured FWHM laser pulse duration was 100 fs (216 fs), the angle of incidence was 25.6° (45.7°), and the beam diameter was 266 μm (≈1 cm) for the pump (probe). The pump beam is slightly elliptical, being around 8% larger along the horizontal axis (266 μm) compared to the vertical axis (220 μm). A quarter-wave plate (QWP) was used for the complex MOKE analysis described in the Supplementary Methods and improving previous methods[30,31,45–47], and a filter was used to prevent the pump beam to reach the CCD camera. A constant external magnetic field of 16 mT was always applied in all measurements. This value of the field was chosen to be the minimum value such that no signal is observed at long time delays.

Images are recorded for each pump-probe delay by a CCD camera (EO-5023M MONO USB CCD camera, exposure time 200 ms, 5 frames/s, accumulation of 40 images) and stored on a computer as tiff files. Once an image is stored, the delay line is moved and the next two accumulations provided by the camera are disregarded before the third accumulation is stored. In this way, we ensure that the stored

images reflect solely the conditions at the designated pump-probe delay τ.

## Modeling

The modeling is based on a two-temperature model (2TM) with heat diffusion described by the following system of partial differential equations:

$$\gamma T_e \frac{\partial T_e}{\partial t} = \frac{\partial}{\partial z}\left(\kappa_e \frac{T_e}{T_p}\frac{\partial T_e}{\partial z}\right) - g_{e-p}\left(T_e - T_p\right) + a(z,t)S_z^0 \qquad (7)$$

$$C_p \frac{\partial T_p}{\partial t} = \kappa_p \frac{\partial^2 T_p}{\partial z^2} + g_{e-p}\left(T_e - T_p\right) \qquad (8)$$

Where $T_e$ is the temperature of the electronic reservoir, $T_p$ is the temperature of the phonon reservoir, the electronic heat capacity $C_e$ (per unit volume) is given by the $\gamma$ parameter as usual via $C_e = \gamma T_e$, $C_p$ is the phonon heat capacity, $\kappa_e$ is the equilibrium electronic heat conductivity (the electronic heat conductivity that would be measured if the electrons were at equilibrium with the phonons), $\kappa_p$ is the phonon heat conductivity, and $g_{e-p}$ is the electron-phonon coupling constant.

The term $a(z,t)S_z^0$ describes the time- and depth-dependent light absorption based on an extension[48] of the transfer matrix method (TMM)[49], which takes into account chromatic dispersion and the real time dependence of Poynting's vector. The electric field **E** with polarization axis **e** at any point of space $(r,z)$ and time $t$ is given by:

$$\mathbf{E}(r,z,t) = E_0(r,z)\mathbf{e}\sum_n G(\omega_n)\exp\left(i(k_n z - \omega_n t)\right) \qquad (9)$$

Where $r$ is the distance from the center axis of the laser beam, $z$ is the distance from the air/sample interface along the center axis of the laser beam, $E_0$ is the electric field amplitude at position $(r,z)$ given by the Gaussian nature of the beam[50], $G(\omega_n)$ gives the Gaussian laser pulse spectrum where in the modeling, one considers a finite set of frequencies $\omega_n$ indexed by $n$, $k_n$ is the wave vector of mode $n$ at position $z$ as given by the standard TMM. The standard deviation $\sigma_\omega$ of the Gaussian spectrum is chosen such that one obtains the desired laser (intensity) pulse duration $\tau_l = 2\sqrt{\ln(2)}/\sigma_\omega$. The electric field calculated in this way is a periodic repetition of pulses. The density of modes, separated by a frequency $\Delta\omega$, is chosen such that the length $2L = 2\pi c/\Delta\omega$ between two laser pulses is 20 times the length of the laser (electric field amplitude) pulse $c2\sqrt{2\ln(2)}/\sigma_\omega$. Poynting's vector is obtained from $\mathbf{S}(r,t) = \mathbf{E}\times\mathbf{H}$ which considers all possible fast oscillations in the energy flow and is not averaged. The normalized "absorption" is defined as:

$$a(z,t) = -\frac{\partial_z S_z}{S_z^0}(z,t) \qquad (10)$$

Where:

$$S_z^0 = \frac{2F\cos(\theta_0)}{\sqrt{2\pi}\sigma_t^l\left(1 + e^{-2(\sigma_t^l\omega_c)^2}\right)} \qquad (11)$$

Is the total energy density flow at the air/sample interface with $F$ the laser fluence, $\theta_0$ the laser pulse angle of incidence at the air/sample interface, $\sigma_t^l = 1/(\sqrt{2}\sigma_\omega)$ and $\omega_c = 2\pi c/\lambda_c$ where $\lambda_c$ is the central wavelength of the laser pulse. The calculation of the absorption relies on the fact that the propagating media are linear and the electric field at any point in time and space can simply be obtained by the sum of plane wave above. One needs the complex optical index of each media for all frequencies $\omega_n$. In the case where the optical indices are frequency

independent and for $\sigma_t^l\omega_c \ll 1$, one retrieves the standard TMM result. The instant $t = 0$ is defined when the center of the laser pulse reaches the air/sample interface ($z = 0$). The absorption is computed in this way for a time interval of four times the laser (electric field amplitude) pulse duration centered on $t = 0$ and is set to zero otherwise.

The ultrafast magnetization dynamics is calculated from the theory in ref. [20] in the special case of a spin ½ ferromagnet:

$$\frac{dm}{dt} = \frac{1}{\tau}\left(m - \frac{\Delta\mu}{2k_BT_C}\right)\left[1 - m\coth\left(\frac{2mk_BT_C - \Delta\mu}{2k_BT_e}\right)\right] \qquad (12)$$

Where $\tau$ is a characteristic time for the transfer of angular momentum between the localized and the conduction electrons, $m$ is the magnetic moment of the localized electrons normalized by its value at zero Kelvin, $T_C$ is the Curie temperature of the ferromagnetic, and $\Delta\mu$ is the spin accumulation in the ferromagnet. The electronic temperature is taken from the numerical solution of the 2TM. Because the model for magnetization dynamics considers systems homogeneous in space, the electronic temperature used as an input in Eq. (12) is the calculated electronic temperature of the middle of the ferromagnetic multilayer. The conduction electrons also contribute to the total magnetic moment $m_{tot} = m - \rho\Delta\mu$, where the form of $\rho$ depends on the details of the model[20,38] and is taken here as a parameter. Because of the thin copper spacer, the spin accumulation generated in GdFeCo is assumed to be instantaneously transferred to the [Co/Pt] multilayer and is obtained from[47]:

$$\Delta\mu = \frac{\tau_s V_{at}}{\bar{D}\mu_B}\frac{dM_{GdFeCo}}{dt} \qquad (13)$$

Where $\tau_s$ is a characteristic time, which must be smaller than the characteristic demagnetization time of GdFeCo[47], treated as a parameter, $V_{at}$ is the average volume of an atomic cell, $\bar{D} = D_\uparrow D_\downarrow/(D_\uparrow + D_\downarrow)$ with $D_{\uparrow(\downarrow)}$ the density of state per unit atom for electrons with up (down) spin and $M_{GdFeCo}$ is the magnetization, all quantities referring to GdFeCo. Sommerfeld's model is used to estimate $\bar{D}/V_{at} = 3\gamma_{GdFeCo}/(\pi k_B)^2$. A more detailed discussion about the generation of spin currents and spin accumulations in nonmagnetic metals in contact with ferromagnets can be found in the work of ref. [36]. For our qualitative model, we neglect the spin accumulation generated by the [Co/Pt] multilayer. The magnetization dynamics of each sublattice of GdFeCo is obtained from the fit obtained in ref. [27] and assumed to be the same for the alloy concentration used in this work. The initial magnetization of each sublattice is adjusted such as to match the equilibrium magnetization of GdFeCo at room temperature ($RT = 300$ K) taken from the work of ref. [51]. The dynamics is taken to be constant at negative time delay before being convolved with a Gaussian function with full width at half maximum given by $\tau_l$. The amplitude of the demagnetization is assumed to scale linearly from 0 to 100% when the fluence is changed from 0 to 5.39 mJ/cm² (the experimental threshold fluence for GdFeCo switching). This linear scaling assumption is incorrect, especially close to the threshold fluence for GdFeCo switching, but it will not change our results qualitatively. The magnetization recovery characteristic times are given by the reversal characteristic times of ref. [27] which is not a quantitatively correct estimation for fluences close to the threshold fluence for GdFeCo switching. This is however unimportant for the fluences required to reverse the magnetization of the ferromagnetic multilayer. P and AP configurations are modeled by considering two different signs of the spin accumulation: the spin accumulation $\Delta\mu$ as plotted in Supplementary Fig. 11a is used as an input in Eq. (12) for simulations in the AP configurations while the opposite $-\Delta\mu$ is used for simulations in the P configuration. The apparent sign discrepancy comes from the fact that the normalized magnetization $m$ is related to spin angular momentum via $m = -\langle S^z\rangle/S$ where $S = 1/2$ in our case and $\langle S^z\rangle$ is the

average of the longitudinal spin angular momentum operator. See Supplementary Methods for more details.

## Data availability
Data shown in this work are available from the corresponding author upon request.

## Code availability
Code used in this work for image analysis and modeling is available from the corresponding author upon request.

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

## Acknowledgements

Q.R. thanks Sébastien Petit-Watelot, Maarten Beens, and Bert Koopmans for helpful discussions. This work is supported by the ANR-20-CE09-0013 UFO, by the Institute Carnot ICEEL for the project "CAPMAT" and FASTNESS, by the Région Grand Est, by the Metropole Grand Nancy, for the Chaire PLUS by the impact project LUE-N4S, part of the French PIA project "Lorraine Université d'Excellence" reference ANR-15-IDEX-04-LUE, by the "FEDERFSE Lorraine et Massif Vosges 2014–2020", a European Union Program, by the European Union's Horizon 2020 research and innovation program COMRAD under the Marie Skłodowska-Curie grant agreement No 861300 and by the Academy of Finland (Grant No. 316857). This article is based upon work from COST Action CA17123 MAGNETOFON, supported by COST (European Cooperation in Science and Technology). All fundings were shared equally among all authors.

## Author contributions

Q.R., G.M., S.M., and M.H. conceived the project. M.H. prepared the samples. Q.R. performed static measurements and subsequent analysis. J.H. designed and built the TR-MOKE microscope with the help of M.V. and J.G. Q.R., J.H., M.V., and Y.L. performed TR-MOKE microscopy measurements and subsequent analysis with the guidance of J.G. and G.M. Q.R. performed simulations. Q.R., J.H., and S.M. wrote the original manuscript with inputs from all authors. All authors discussed the results. G.M., S.M., and M.H. supervised the project.

## Competing interests

The authors declare no competing interests.
