## [Peer Review File · Nature Communications]

Reviewers' Comments:

Reviewer #1:

Remarks to the Author:

I very much appreciate the care of the authors with which they have addressed the issues raised by the referees. My questions have been mostly satisfactorily answered and I agree that the changed manuscript has greatly improved. I repeat my statement from my first report that I am convinced that the presented concept of overcoming CSD is novel and of high significance, even for potential future applications.

My only remaining concern is that the main manuscript still heavily relies on the very extensive extended Method and Supplementary section as well as previous published results. This makes it somewhat tedious to read. (As a side note, the preprint view with figures shifted to the back of the paper is really a bit of an imposition for referees and actually quite annoying, as it requires displaying at least 3 copies of the pdf and still scrolling back and forth all the time).

On a positive note, I felt the new introduction is very well written and a very good and general description of the problem of CSD. Its generality is appreciated when the authors refer to other examples, in particular to ferromagnetic systems (Ref 27-29): for AOS in GdFe the ΔE_{ext} is provided by the Gd sublattice when m approaches zero.

Regarding the re-switching times of 650 fs, I think it is important to clearly differentiate this from double pulse excitation experiments. With a double pulse experiment, one has significantly more control over the times and more importantly, each pulse leads to a well defined final state. While I agree that in principle there are also two stimuli (optical light and spin current), here the switched state is only a transient state? How would this compare to the results of Alebrand et al. PRB 89, 144404 (2014), DOI: 10.1103/PhysRevB.89.144404, Fig. 2, which shows a transient switched state in TbCo on fast time scales?

Line 104 ff: I am now a bit confused regarding the geometry of the excitation scheme. Which is side is excited, which one is probed? In the text it says that the (pinned) magnetic [Co(1)/Pt(1)]₂/Co(0.6)/GdFeCo(5) system is the spin source, this is why I had previously thought that this is the layer that is excited, similarly as in reference 33 (Iihama et al.). But it is not explicitly mentioned anywhere in the manuscript. Also, in the Extended Figure 3a it seems different: here the free layer is excited. Also in Extended Figure 2a the pump and probe pulse impinge from the same side Co/Pt, but is the microscopy not done from the backside, as shown in Extended Figure 3a?

Line 150: Equation (5), please define $\Delta \mu$ (only defined much later, line 464 as spin accumulation). A few words on what is meant by this would be appreciated. Currently the understanding relies on (re-)reading the work of Beens et al..

Fig. 3b): Why is m_{tot} in the calculation not 1 or -1 before $t = 0$?

While I appreciate the details given in section "Modelling", it seems to contain so many simplifications, assumptions and free (temperature dependent?) parameters that it become difficult to judge whether it can really be considered as a robust description of the observed dynamics. More specifically for example:

Line 479: The magnetization dynamics is taken from data by Radu et al., but is the dynamics not strongly dependent on the excitation fluence?

Line 484: Does the magnetization really scale linearly with fluence up to the switching threshold? It is also a bit unfortunate that the transiently switched state cannot be reproduced, it may be helpful to briefly discuss the most important assumptions and free parameters and their effect on the results.

"Pump-probe delay" is not consistently written with a hyphen.

Reviewer #2:

Remarks to the Author:

Manuscript NCOMMS-22-27243-T entitled 'Accelerating ultrafast magnetization reversal by non-local spin transfer' by authors Quentin Remy et al. reports on ultrafast reversal of magnetization observed in all-optical time-resolved Kerr microscopy. The authors interpret their data using a two-temperature model featuring additional angular momentum from spin accumulation generated by an ultrafast laser pulse. The work demonstrates a methodology to achieve an ultrafast reversal using spin heating and spin cooling mechanisms to overcome the relatively slow limitation of critical slowing down of the magnetization reversal associated with a phase transition to a paramagnetic state.

The manuscript is well written and quite well organised and supplemental information is provided for further details of the experiment and model. The findings will be of immediate interest to researchers in the field of ultrafast magnetization dynamics, but will be of broader interest to those working in ultrafast non-linear processes and processes far from equilibrium, and perhaps those developing magnetic data storage technologies where rapid switching between bistable states is necessary.

The manuscript should be publishable in Nature Communications but there are a few minor issues that should be addressed for the benefit of the reader before the manuscript is published.

As I understand the authors claim that full reversal to the full magnetization is not seen because 1. the material has fairly low T_c and is still hot following the ultrafast heating and so the reversed magnetization is small at several ps, and 2. at longer timescales it does not relax to the switched state because of a static reset field that pulls the magnetization back to the original state. It would be useful if the authors could state explicitly show which states the dynamic process moves between on the hysteresis loop. From the hysteresis loop in extended figure 2b it seems that the applied field for the switched state exceeds the static switching field for the CoPt layer so it will slowly reverse back to the original state.

On page 6 the authors write 'At first, the ultrafast demagnetization is shortly slowed down...'. It is not clear what this means, do the authors mean rapidly slowed down, or slows down after a short time delay?

In figure 2 could the optical pump duration somehow be shown as I assume it is not instantaneous as indicated but perhaps ~ 50 to 100 fs or so.

On page 10, the following statement was confusing. 'Indeed, as shown on the inset of Figure 1, if magnetization crosses zero (from a positive value), a further decrease of magnetization (towards negative values) will result in a spin cooling.' Firstly is Figure 1 the correct figure? Is this statement describing the schematic diagram of Figure 1? It is also not clear which parts of Figure 1 are the referenced inset. Secondly, if the magnetization 'crosses zero (from a positive value)' the magnetization is momentarily zero, presumably due to ultrafast heating and demagnetization, so then the material begins to remagnetise (with increasing magnetization) in the negative direction, so perhaps it does not exhibit a 'further decrease of magnetization towards negative values'. Can the authors check the phrasing of this statement to make sure it is clear for the reader.

In the Methods section under Static Measurements the authors write 'Hysteresis loops have been measured with the TR-MOKE microscopy setup as shown in Figure 2.' Figure 2 does not show the TR-MOKE microscopy set up, perhaps it should be extended Figure 5? Also state that the TR capability is not required for the loops and the pump was blocked for the loop measurement (I assume).

Please check this sentence in the following paragraph. 'The threshold fluences required to reverse a given magnetic field where...' I think the authors mean 'reverse the magnetization at a given magnetic field where...'

Extended Data Fig 3a. Is the schematic shown a different optical pump-probe configuration to that shown in extended data figure 2a? In 2a the pump and probe are incident on the CoPt, in 3a they are incident from opposite sides with the pump on the GeFeCo side of the stack. In 3a authors have also flipped the stack configuration compared to Fig 2a. The authors should consider making the two figures consistent in terms of stack orientation, or even removing one of the figures to avoid confusion for the reader, since they seem to show almost the same information. If there are intended differences, please explicitly state why in each caption. E.g. why is it necessary in 2a to show a pump when a static hysteresis loop is shown, was the pump applied when the loop was measured? If not then write e.g. hysteresis loop acquired with no pump present. In these two cases why was the pump from different sides of the stack. Maybe I missed the reasoning, but it was not clear to me from the manuscript of extended data.

Dear reviewers,

We thank the reviewers for their review of our manuscript "Accelerating ultrafast magnetization reversal by non-local spin transfer".

We address below all the comments of the reviewers (blue and italic).

Reviewer #1 (Remarks to the Author):

I very much appreciate the care of the authors with which they have addressed the issues raised by the referees. My questions have been mostly satisfactorily answered and I agree that the changed manuscript has greatly improved. I repeat my statement from my first report that I am convinced that the presented concept of overcoming CSD is novel and of high significance, even for potential future applications.

We thank the reviewer for their positive comments and we appreciated previous comments which we indeed found very helpful to improve our article.

My only remaining concern is that the main manuscript still heavily relies on the very extensive extended Method and Supplementary section as well as previous published results. This makes it somewhat tedious to read. (As a side note, the preprint view with figures shifted to the back of the paper is really a bit of an imposition for referees and actually quite annoying, as it requires displaying at least 3 copies of the pdf and still scrolling back and forth all the time).

We agree with the reviewer that the Methods and Supplementary sections are quite long. However, we believe that all the fundamental information can be found in the main text and we use the Methods and Supplementary to address more technical aspects which enable the reproducibility of our measurements and simulations.

The first major part of the Methods and Supplementary is about our time resolved (complex) magneto optical Kerr effect (MOKE) microscopy. This part is very important because all our work relies on the fact that we can measure separately the magnetization dynamics of both magnetic layers and that the zero crossing (at very short timescale) that we observe is not due to some kind of artifact. Several previous studies have used complex MOKE but it has been reported (see for instance reference 47) that one is always sensitive to both layers in practice and we had to circumvent this problem. The method to obtain the magnetization dynamics of a single layer used in the submitted article has never been published before, as far as we know.

The second major part of the Methods and Supplementary focuses on the simulations (which are also addressed below). The method used has also never been introduced before, even though it is based on existing physics.

On a positive note, I felt the new introduction is very well written and a very good and general description of the problem of CSD. Its generality is appreciated when the authors refer to other examples, in particular to ferromagnetic systems (Ref 27-29): for AOS in GdFe the ΔE_{ext} is provided by the Gd sublattice when m approaches zero.

Regarding the re-switching times of 650 fs, I think it is important to clearly differentiate this from double pulse excitation experiments. With a double pulse experiment, one has significantly more control over

the times and more importantly, each pulse leads to a well defined final state. While I agree that in principle there are also two stimuli (optical light and spin current), here the switched state is only a transient state? How would this compare to the results of Alebrand et al. PRB 89, 144404 (2014), DOI: 10.1103/PhysRevB.89.144404, Fig. 2, which shows a transient switched state in TbCo on fast time scales?

We agree that double optical pulse excitation experiments are different. To make sure that this is clearly explained in our manuscript, at the end of the Subpicosecond magnetization reversal section, we added : " We consider this magnetization reswitching or rewriting, **caused at high temperatures by a fast sign change of the spin current that originates from heating by a single laser pulse,** in the next section."

The switched state is indeed transient, we argue that it is not an issue for practical applications since, in the case where the second peak of the spin current did not exist, magnetization would eventually reach equilibrium (in the switched state). The important part is that we have very little limitations for the speed at which the state of magnetization is set. The only fundamental limitations are the speed of ultrafast demagnetization and the speed of the angular momentum transfer (between s and d electrons in our simplified picture). For practical applications, one then just needs to be able to engineer the shape of spin currents, as magnetization is fully controlled by the spin current when the magnetization is sufficiently close to zero. Then information can be permanently stored in a few hundreds of femtoseconds by sending the right amount of energy and angular momentum, and the system is already ready to be reversed again.

Regarding the results of Alebrand and coworkers, we assume the reviewer is talking about Fig 2 b for 32% of Tb. We note that these results are rather different as in their case, magnetization crosses zero once before going to zero on longer timescales, while in our case, magnetization crosses zero a second time before recovering. Also, as the authors remark, the signal measured with a wavelength of 400 nm is a superposition of contributions from both magnetic sublattices, even though it is dominated by the Tb sublattice (note also the assumptions made in the discussion section of this article). Overall, we thus believe that both transient states (in their work and ours) are quite different.

Line 104 ff: I am now a bit confused regarding the geometry of the excitation scheme. Which is side is excited, which one is probed? In the text it says that the (pinned) magnetic [Co(1)/Pt(1)]₂/Co(0.6)/GdFeCo(5) system is the spin source, this is why I had previously thought that this is the layer that is excited, similarly as in reference 33 (Iihama et al.). But it is not explicitly mentioned anywhere in the manuscript. Also, in the Extended Figure 3a it seems different: here the free layer is excited. Also in Extended Figure 2a the pump and probe pulse impinge from the same side Co/Pt, but is the microscopy not done from the backside, as shown in Extended Figure 3a?

In all the time resolved experiments (Extended Data Fig 2), the sample is pumped and probed from the sample side i.e. the [Co(0.6)/Pt(1)]₂ multilayer (reversing) of interest. In the static microscopy measurements (Extended Data Fig 3), the sample is pumped from the sample side as well while microscopy (observation of the magnetic state) is performed from the substrate side (i.e. directly exciting the [Co(1)/Pt(1)]₂/Co(0.6)/GdFeCo(5) multilayer). Thus, in both cases, the sample is pumped from the sample side, directly exciting the [Co(0.6)/Pt(1)]₂ multilayer. This was required to limit the illumination of the [Co(1)/Pt(1)]₂/Co(0.6)/GdFeCo(5) multilayer and demagnetize the [Co(0.6)/Pt(1)]₂ multilayer more easily. Otherwise higher fluences were required for the switching which could cause damage of the [Co(1)/Pt(1)]₂/Co(0.6)/GdFeCo(5) multilayer after several hours of exposure to laser pulses for the pump-probe measurements. Probe/microscopy were performed on different sides for static and dynamic measurements because two different experimental setups have been used for both types of measurements. The experimental setup used to perform static measurements was described in reference 35 as mentioned in the Methods section.

To make this clearer in the text, we added, in the Static measurements section of the Methods: " In these measurements, a 35 fs linearly polarized laser pulse (800 nm) is irradiated on the top [Co(0.6)/Pt(1)]₂ side of the sample while static (not time resolved) MOKE microscopy is performed on the other side of the sample." and in the TR-MOKE microscopy section of the Methods: " Both the pump and the probe pulses irradiate the top [Co(0.6)/Pt(1)]₂ side of the sample as shown in Extended Data Fig. 2."

Line 150: Equation (5), please define Delta mu (only defined much later, line 464 as spin accumulation). A few words on what is meant by this would be appreciated. Currently the understanding relies on (re-)reading the work of Beens et al..

We thank the reviewer for their comment. We already introduced the spin accumulation earlier but without the notation. In line 99, we added: " when ΔE_{ext} is (minus) the spin accumulation $\Delta\mu$ of conduction electrons²⁰ i.e. the difference between the chemical potential for spin up and spin down electrons."

Fig. 3b): Why is m_{tot} in the calculation not 1 or -1 before $t = 0$?

The normalized magnetization in these simulations is customarily normalized with respect to the saturated magnetization $m = -\langle S_z \rangle / S$ ($\langle \cdot \rangle$ denotes the standard thermal average of an operator) i.e. magnetization at zero temperature, which is also what we have done here. Because we perform simulations when the initial temperature is non zero, $m_{\text{tot}}(t < 0) = m(t < 0) - \rho \cdot \Delta\mu(t < 0) = m(t < 0)$ will be different from 1 or -1.

While I appreciate the details given in section "Modelling", it seems to contain so many simplifications, assumptions and free (temperature dependent?) parameters that it become difficult to judge whether it can really be considered as a robust description of the observed dynamics. More specifically for example:

We agree that our model contains many simplifications and assumptions. However, most of the parameters were directly taken from the literature. Only the Curie temperature of Co/Pt ($T_C = 500\text{K}$) and the spin relaxation time $\tau_s = 35\text{fs}$ (used to estimate the amplitude of the spin accumulation) were used. Both values are reasonable compared to what is discussed in the literature.

Our aim with these simulations is not to quantitatively reproduce the experiments. No fitting is done and we are well aware that the dynamics of such system is a very complex problem.

The essential physics is contained in Equation (5): one needs enough heat to demagnetize the system; then m reaches 0 (and $\Delta\mu$ becomes larger than $2m k_B T_C$) and will evolve according to the amplitude of the spin accumulation $\Delta\mu$.

Line 479: The magnetization dynamics is taken from data by Radu et al., but is the dynamics not strongly dependent on the excitation fluence?

The dynamics is indeed strongly dependent on fluence (especially close to the threshold fluence for magnetization reversal). We note however that the ultrafast demagnetization (which contributes the most to the generation of the spin accumulation) does not change much with fluence (contrary to the magnetization dynamics during recovery or after/during reversal).

Line 484: Does the magnetization really scale linearly with fluence up to the switching threshold?

It will not scale linearly with fluence but we would then need an accurate description of the GdFeCo dynamics (possibly coupled to its Co/Pt multilayer) which would inevitably introduce further parameters. Moreover, we do not expect that this would change the qualitative results of the model, thus we chose to keep the simplest model.

To clarify this point, we added in the text of the Methods section: "This linear scaling assumption is incorrect, especially close to the threshold fluence for GdFeCo switching, but it will not change our results qualitatively."

It is also a bit unfortunate that the transiently switched state cannot be reproduced, it may be helpful to briefly discuss the most important assumptions and free parameters and their effect on the results.

A transient switched state could be reproduced with different parameters. However, we did not find another set of parameters which could reproduce all the qualitative features of our experiments at the same time (three step dynamics in the P configuration, transient reversal in the AP configuration, similar threshold fluences, similar reversed amplitude).

All parameters are temperature independent. All temperature dependences are explicitly written.

All the assumptions are discussed in the Methods. We note that a discussion of the effect of the free parameters was already done at the end of the Supplementary Methods. We extended it to discuss the impact of the Curie temperature: "The value of τ_s is one of the most important parameter of these simulations as it gives the amplitude of the spin accumulation. The small value that we used is reasonable considering typical spin relaxation times in ferrimagnets³² and ferromagnets⁵¹ below 0.1 ps. An order of magnitude smaller values of τ_s would be unrealistic. Higher values increase the magnetization reversal speed and reduce its required threshold fluence. The value of the parameter ρ does not influence the energy efficiency or speed of the reversal in this model but only affects the (transient) contribution of the spin accumulation to the magnetization whose value is much smaller than the equilibrium magnetization of the ferromagnetic multilayer. Decreasing T_c results in a smaller reversed amplitude (thus a better match with the experiments on a long timescale) but (i) reduces the threshold fluence and (ii) makes the three step dynamics (ultrafast demagnetization; non equilibrium recovery; equilibrium recovery) less obvious (the dynamics looks like what we obtained at higher fluence in Extended Data Fig. 11c)."

"Pump-probe delay" is not consistently written with a hyphen.

We thank the reviewer for point out this inconsistency. This has been corrected.

Reviewer #2 (Remarks to the Author):

Manuscript NCOMMS-22-27243-T entitled 'Accelerating ultrafast magnetization reversal by non-local spin transfer' by authors Quentin Remy et al. reports on ultrafast reversal of magnetization observed in all-optical time-resolved Kerr microscopy. The authors interpret their data using a two-temperature model featuring additional angular momentum from spin accumulation generated by an ultrafast laser pulse. The work demonstrates a methodology to achieve an ultrafast reversal using spin heating and spin cooling mechanisms to overcome the relatively slow limitation of critical slowing down of the magnetization reversal associated with a phase transition to a paramagnetic state.

The manuscript is well written and quite well organised and supplemental information is provided for further details of the experiment and model. The findings will be of immediate interest to researchers in the field of ultrafast magnetization dynamics, but will be of broader interest to those working in ultrafast non-linear processes and processes far from equilibrium, and perhaps those developing magnetic data storage technologies where rapid switching between bistable states is necessary.

The manuscript should be publishable in Nature Communications but there are a few minor issues that should be addressed for the benefit of the reader before the manuscript is published.

We thank the reviewer for their positive comments.

As I understand the authors claim that full reversal to the full magnetization is not seen because 1. the material has fairly low T_c and is still hot following the ultrafast heating and so the reversed magnetization is small at several ps, and 2. at longer timescales it does not relax to the switched state because of a static reset field that pulls the magnetization back to the original state. It would be useful if the authors could state explicitly show which states the dynamic process moves between on the hysteresis loop. From the hysteresis loop in extended figure 2b it seems that the applied field for the switched state exceeds the static switching field for the CoPt layer so it will slowly reverse back to the original state.

We fully agree with the reviewer's comment. As proposed by the reviewer, Extended Data Fig. 2 was changed to indicate that magnetization goes from P+ to AP- (or P- to AP+) upon reversal of the Co/Pt multilayer. Accordingly, we added in the caption: "The thin blue arrows in the minor hysteresis loops indicate the change of MOKE signal upon reversal of the (top) single [Co/Pt] multilayer."

The applied field was chosen such as to satisfy two conditions: (i) there is no signal at long time delays (i.e. at negative time delays) so that magnetization is reinitialized before each subsequent pump pulse and (ii) the field is not too high so that it has a minimum impact on the observed magnetization dynamics. So indeed, the Co/Pt layer will slowly reverse back to the original state but fast enough to be completely reset before the next pump-pulse as required in pump-probe measurements.

On page 6 the authors write 'At first, the ultrafast demagnetization is shortly slowed down...'. It is not clear what this means, do the authors mean rapidly slowed down, or slows down after a short time delay?

We meant "slowed down for a short period of time" i.e. during the period of time where the Co/Pt multilayer receives the spin accumulation due to the FeCo sublattice.

Line 128 now reads: "At first, the ultrafast demagnetization is slowed down for a short period of time due to ultrafast spin cooling"

In figure 2 could the optical pump duration somehow be shown as I assume it is not instantaneous as indicated but perhaps ~50 to 100 fs or so.

We assume the reviewer refers to Figure 2a for zero time delay. We added a gaussian curve on the time axis to represent the pulse duration. Due to lack of space, we added "The red shaded gaussian curve represents the 100 fs (800 nm) pump laser pulse." in the caption of the figure.

On page 10, the following statement was confusing. 'Indeed, as shown on the inset of Figure 1, if magnetization crosses zero (from a positive value), a further decrease of magnetization (towards negative values) will result in a spin cooling.' Firstly is Figure 1 the correct figure? Is this statement describing the schematic diagram of Figure 1? It is also not clear which parts of Figure 1 are the referenced inset. Secondly, if the magnetization 'crosses zero (from a positive value)' the magnetization is momentarily zero, presumably due to ultrafast heating and demagnetization, so then the material begins to remagnetise (with increasing magnetization) in the negative direction, so perhaps it does not exhibit a 'further decrease of magnetization towards negative values'. Can the authors check the phrasing of this statement to make sure it is clear for the reader.

We are talking about the T_S vs m graph, in the top left corner of in Figure 1. We changed "Indeed, as shown on the inset of Figure 1" to "Indeed, as shown in the top left corner of Figure 1"

What we mean is that when m goes from 0 to -1, it decreases because we do not talk about the absolute value of magnetization but rather its projection along an axis. To make it simpler, we changed "if magnetization crosses zero (from a positive value), a further decrease of magnetization (towards negative values) will result in a spin cooling." to " **spin heating becomes spin cooling when magnetization crosses zero**"

In the Methods section under Static Measurements the authors write 'Hysteresis loops have been measured with the TR-MOKE microscopy setup as shown in Figure 2.' Figure 2 does not show the TR-MOKE microscopy set up, perhaps it should be extended Figure 5? Also state that the TR capability is not required for the loops and the pump was blocked for the loop measurement (I assume).

We thank the reviewer for pointing out this error. It is indeed Extended Data Fig 5. It now reads: " Hysteresis loops have been measured with the TR-MOKE microscopy setup as shown in **Extended Data Fig. 5 with the pump blocked.**"

Please check this sentence in the following paragraph. 'The threshold fluences required to reverse a given magnetic field where...' I think the authors mean 'reverse the magnetization at a given magnetic field were...'

We thank the reviewer for pointing out this error. What we meant was : " The threshold fluences required to reverse a given magnetic **domain** where obtained following the method".

In the text, " The threshold fluences required to reverse a given magnetic **field** where obtained following the method" was changed to " The threshold fluences required to reverse a given magnetic **domain** where obtained following the method".

Extended Data Fig 3a. Is the schematic shown a different optical pump-probe configuration to that shown in extended data figure 2a? In 2a the pump and probe are incident on the CoPt, in 3a they are incident from opposite sides with the pump on the GeFeCo side of the stack. In 3a authors have also flipped the stack configuration compared to Fig 2a. The authors should consider making the two figures consistent in terms of stack orientation, or even removing one of the figures to avoid confusion for the reader, since they seem to show almost the same information. If there are intended differences, please explicitly state why in each caption. E.g. why is it necessary in 2a to show a pump when a static hysteresis loop is shown, was the pump applied when the loop was measured? If not then write e.g. hysteresis loop acquired with no pump present. In these two cases why was the pump from different sides of the stack. Maybe I missed the reasoning, but it was not clear to me from the manuscript of extended data.

The sketch shown in Extended Data Fig 3a shows the experimental configuration for the estimation of threshold fluences as described in reference 35. In this case, we do not perform a pump-probe measurement but only standard MOKE microscopy after sending a pulse. As in the time resolved measurement, a 800 nm pulse is send from the sample side (reversed Co/Pt side) but (static) microscopy is performed on the other side for this different experimental setup.

We changed the order of the stack in Extended Data Fig. 2a to be consistent with Extended Data Fig 3a.

We emphasized in the caption of Extended Data Fig. 3 that the experimental configuration shown in Extended Data Fig 3a corresponds to static measurements: " The laser pulse irradiates the top side of the sample **in** standard **(static)** MOKE microscopy".

We indicated in the caption of Extended Data Fig. 3b that the pump laser was blocked to measure the hysteresis loop: " Typical major (black) and minor (gray) hysteresis loops measured via the Kerr rotation, measured on our TR-MOKE microscope, as function of a perpendicular external field, **when the pump laser is blocked.**".

Reviewers' Comments:

Reviewer #1:

Remarks to the Author:

All my questions and concerns have been satisfactorily answered.

Reviewer #2:

Remarks to the Author:

The authors of manuscript NCOMMS-22-27243-T entitled 'Accelerating ultrafast magnetization reversal by non-local spin transfer' have address all questions and comments raised in the earlier review. I am satisfied with the author's response to my comments (and to those of the first reviewer), and with the corresponding amendments to the manuscript. My recommendation is to publish the manuscript in Nature Communications.

Dear reviewers,

We thank the reviewers for their review of our manuscript "Accelerating ultrafast magnetization reversal by non-local spin transfer".

We address below all the comments of the reviewers (blue and italic).

Reviewer #1 (Remarks to the Author):

All my questions and concerns have been satisfactorily answered.

We thank the reviewer for the positive review of our manuscript.

Reviewer #2 (Remarks to the Author):

The authors of manuscript NCOMMS-22-27243-T entitled 'Accelerating ultrafast magnetization reversal by non-local spin transfer' have address all questions and comments raised in the earlier review. I am satisfied with the author's response to my comments (and to those of the first reviewer), and with the corresponding amendments to the manuscript. My recommendation is to publish the manuscript in Nature Communications.

We thank the reviewer for the positive review of our manuscript.